# Multi-Scale Surface Treatments of Titanium Implants for Rapid Osseointegration: A Review

**DOI:** 10.3390/nano10061244

**Published:** 2020-06-26

**Authors:** Qingge Wang, Peng Zhou, Shifeng Liu, Shokouh Attarilar, Robin Lok-Wang Ma, Yinsheng Zhong, Liqiang Wang

**Affiliations:** 1School of Metallurgical Engineering, Xi’an University of Architecture and Technology, No.13 Yanta Road, Xi’an 710055, China; wendymewqg@163.com; 2School of Aeronautical Materials Engineering, Xi’an Aeronautical Polytechnic Institute, Xi’an 710089, China; pzhou1975@163.com; 3State Key Laboratory of Metal Matrix Composites, School of Material Science and Engineering, Shanghai Jiao Tong University, Shanghai 200240, China; sh.attarilar@yahoo.com; 4Department of Mechanical and Aerospace Engineering, The Hong Kong University of Science and Technology, Hong Kong 999077, China; melwma@ust.hk (R.L.-W.M.); rinto@connect.ust.hk (Y.Z.); 5National Engineering Research Center for Nanotechnology (NERCN), 28 East JiangChuan Road, Shanghai 200241, China

**Keywords:** macro-scale, micro-scale, nano-scale, surface modification, roughness, rapid bone integration

## Abstract

The propose of this review was to summarize the advances in multi-scale surface technology of titanium implants to accelerate the osseointegration process. The several multi-scaled methods used for improving wettability, roughness, and bioactivity of implant surfaces are reviewed. In addition, macro-scale methods (e.g., 3D printing (3DP) and laser surface texturing (LST)), micro-scale (e.g., grit-blasting, acid-etching, and Sand-blasted, Large-grit, and Acid-etching (SLA)) and nano-scale methods (e.g., plasma-spraying and anodization) are also discussed, and these surfaces are known to have favorable properties in clinical applications. Functionalized coatings with organic and non-organic loadings suggest good prospects for the future of modern biotechnology. Nevertheless, because of high cost and low clinical validation, these partial coatings have not been commercially available so far. A large number of in vitro and in vivo investigations are necessary in order to obtain in-depth exploration about the efficiency of functional implant surfaces. The prospective titanium implants should possess the optimum chemistry, bionic characteristics, and standardized modern topographies to achieve rapid osseointegration.

## 1. Introduction

In recent decades, the worldwide demand for dental and orthopedic implants has grown steadily, reaching approximately $45.5 billion sales in 2014 [1,2,3]. Brånemark [4] studied the osseointegration process and applied the first dental implant in the 1960s. Since then, the detailed study and development of dental and orthopedic implants have been continued. Long-term follow-up for the different types of implants in patients has been adequately reported in the literature [5,6,7,8,9,10]. The clinical success rate of dental implants was reported to be more than 87.8% over a follow-up period of 36 years, which is mainly related to early bone regeneration [5]. Dental implant design and its topography are among the vital factors influencing its early osseointegration process. Since the 1970s, dental implant shapes have transformed from hexagonal to conical connections and they are usually designed as rough titanium surfaces [1,5,11]. The efficiency of the connection method directly affects the long-term stability of the bone tissue in the neck of the dental implant. The long-term stability of conical connections is better than that of hexagonal connections. In addition, the rough surface increases the contact area between the implant and the osteoblasts, thus accelerating bone healing. It also reduces bone resorption by increasing the bonding strength and thus improving the interfacial stress distribution, which in turn reduces the healing time in dental implants [11,12]. Two types of bone to implant surface interactions are observed in the initial stages of osseointegration. The first type involves a fibrous soft tissue capsule formation, and if it does not achieve the proper fixation with the surrounding bone, it will probably lead to implant failure. The second type, associated with the direct interaction of bone with the implant surface, is defined as osseointegration [1]. It is generally recognized that high fixation is one of the prerequisite parameters for successful long-term implantation [13]. The rate of osseointegration and the percentage of bone-to-implant contact (BIC) are highly dependent on the surface properties [14,15,16]. Various parameters such as chemical composition, surface energy, wettability (hydrophobicity/hydrophilicity), roughness, topography, and surface morphology play crucial roles in adhesion and the survival of cells [17,18]. Usually, materials with excellent biological and mechanical properties such as commercially pure titanium (CP Ti), Ti-6Al-4V, and zirconia have been previously used in dental and orthopedic implants [19,20,21,22,23,24]. 

Considering the non-toxic nature and biocompatibility characteristics, titanium is one of the best choices in implant applications. Titanium shows a vast number of remarkable properties, for instance, high fatigue and corrosion resistance in biological fluids [25]. Furthermore, among various Ti alloys, the β-type alloys reveal lower elastic moduli [26,27], excellent corrosion resistance [28,29], and improved biocompatibility [30,31,32,33,34,35,36,37,38,39]. However, the bio-inertness of Ti alloys leads to an extended osseointegration time with bone. In order to overcome this limitation, surface treatment technologies can be used to attain bioactive surfaces on Ti substrates [40,41,42,43]. The macro-scale, micro-scale, and nano-scale morphology of the implant surfaces have a crucial influence on the early bone formation and fixation [16,44,45]. Most titanium implant surfaces with certain roughness characteristics were fabricated through mixed technologies (e.g., grit-blasting, acid-etching). In addition, the latest research literature concentrates on macro-, micro-, and nano-scale surface modification through different methods with promoted osseointegration responses [42,46,47]. Meanwhile, the multi-scaled morphologies enhance protein adsorption and stimulate osteogenic cell migration in order to accelerate the osseointegration period [48]. In addition, periodontitis (CP) and coronary heart disease (CHD) patients have bigger challenges regarding dental implant implantation, as these patients are more likely to develop peri-implantitis. In clinical studies, asymmetric dimethylarginine (ADMA) [49], endothelin-1(ET-1) concentrations [50], and vitamin D [51] have nonnegligible effects on CP and CHD. These studies suggested that patients suffering from CP and CHD have higher salivary levels of ET-1 and lower serum levels of vitamin D than healthy control subjects [50,51]. In a multivariate model, the significant predictors of salivary ADMA levels were hs-C-reactive protein [49]. Therefore, the exact role of the potential benefits of ADMA, ET-1, and vitamin D should be further studied in detail.

The purpose of this article is to report the state of the art on the multi-scale technological advancements of titanium implant surfaces to accelerate osseointegration. This review mainly focuses on innovative physicochemical procedures in multi-scale-based techniques. The physical and chemical characteristics such as wettability, roughness, and bioactivity of titanium implants in relation to biological performance is fully discussed. In this regard, the multi-scale functional coatings have the potential to increase the protein adsorption and speed up the osteogenic cell migration, angiogenesis and the early bone formation and its mineralization. Nevertheless, the optimum process parameters for various technologies still need to be clarified and will be discussed in detail in this article.

### 1.1. Chemical Composition and Wettability

In general, CP Ti and its alloys are used to fabricate dental implant fixtures [52]. Meanwhile, the choice of titanium and its alloys as an implant material depends on the high biocompatibility, corrosion resistance, strength, and the osseointegration function [53]. In addition, the biocompatibility of titanium-based alloys is determined by its composition, patient health conditions, and implanting position. Compared with other metallic systems, pure titanium and its alloys are clinically preferred because of their biosafety and the low density value of about 4.51 g/cm^3^ [54]. CP Ti according to its purity, oxygen, carbon, and iron elements is usually classified into four grades (graded from I to IV); this chemical content determines the purity and grade of CP Ti [1,11]. Most of the used implant fixtures are manufactured from CP Ti grade IV and the abutments are made from Ti-6Al-4V alloy (grade V titanium alloy) [1,11]. The yield strength and fatigue properties of Ti-6Al-4V are higher than that of pure titanium, and the annealed Ti–6Al–4V has a yield strength in the range of 825–869 MPa and a plasticity of about 6–10%, to bear the stress magnitude from occlusal loading [55,56,57]. Furthermore, the wettability of titanium implant surfaces affects cell behavior in the initial osseointegration stage [7,17,58,59]. Considering the interaction of human body fluids, cells, and tissues with the implant surface, hydrophilic surfaces (water contact angle is ranging from 40° to 70°) are more suitable than hydrophobic surfaces [7,60]. The optimum parameters and characteristics regarding the contact angle are still controversial. Previous research [61] revealed that a hydrophilic surface optimization using Sand-blasted, Large-grit, and Acid-etching (SLA) led to a higher BIC percentage of 81.91% than a regular SLA process with 66.57% on CP Ti surfaces four weeks after implanting in miniature pigs. 

### 1.2. Roughness and Morphology 

The human skeleton exhibits a hierarchical structure of the macro-, micro-, and nano-scale levels, as seen in Figure 1 [62]. Bone consists of organic (type I and type IV collagen and fibrillin) and inorganic mineral (hydroxyapatite, HA) constituents. Considering the bone structure and density, its structure can be classified into two main types of bone: trabecular bone (cancellous bone) and cortical bone (compact bone). Cancellous bone is made of a porous network, and its porosity is in the range of 50–90%, depending on the specific location and age. Compact bone has a compact structure with a porosity in the range of 3–12% [14,63,64].

Numerous studies [17,65,66] indicate that the roughness and morphology features of the implant surface have considerable effects on the osseointegration rate and its fixation quality with bone. Surface roughness can be divided into three levels: macro-scale, micro-scale, and nano-scale [15]. The topographical roughness in macro-scale ranges from millimeters to tens of microns. Most of the macro-scale features are fabricated with screws, modifying the roughness to more than 10 μm [1,67]. The initial fixation and stability of implants can be increased by roughening the smooth surfaces. Furthermore, surfaces with high roughness values lead to a better interlocking reaction in the implant bone interface zone compared to smooth surfaces. Nevertheless, surfaces with high roughness also have some limitations such as the increase of peri-implantitis and ion leakage. The roughness value in the micro-scale condition ranges between 1 and 10 μm; this roughness range represents the best interlocking reaction between mineralized bone and implants [1,65,68]. A study reported [1] that the surface should be fabricated with hemispherical pits with approximately 1.5 μm depth and 4 μm diameter. The nano-scaled topography condition is in the range of less than one micrometer. There are various surface morphologies made using different techniques and the related adjusting process parameters, for instance, nano pit, nanotubular, nanowire, nanorod, and nanopore [44,69,70,71,72,73]. Accordingly, the Three-Dimensional Printing (3DP) and Laser Surface Texturing (LST) procedures improve the surface morphology at the macro-scale, while the grit-blasting and acid-etching procedures can produce surface features and morphologies at the micro-scale [74,75,76,77,78,79,80]. Plasma-spraying and anodization processes modify the morphology at nano-scale [81,82,83].

## 2. Results and Discussion

### 2.1. Macro-Scale Treatment

#### 2.1.1. Three-Dimensional Printing

3DP is a well-established and versatile additive material technology that attracts the researchers’ attention due to its individuation in the fabrication of complex constructs [84]. Nowadays, in 3DP implant preparation, the major focus changes from mechanical strength optimization toward rapid bone regeneration and infection inhibition. 3DP technologies have the capability to fabricate porous implants with precise mechanical properties, favorable pore architectures, and even produce implants with patient-specific functional designs [8,83,85]. The 3DP manufactured implants from titanium and its alloys have been thoroughly studied and clinically used for decades, however, further research in this field is necessary to develop stabilized long-term properties. In this section, the factors influencing bone regeneration (for example, pore size, porosity, pore structure, and roughness) are discussed [86]. Meanwhile, the bone-formation ability of titanium implants by means of different manufacturing techniques will be explained, systematically [87,88]. It should be considered that the material structure design combined with biomimetic functionalization in order to enhance its long-term osseointegration capacity is necessary.

Over the past few decades, varieties of 3DP techniques have been thoroughly studied [8,89,90,91]. It is generally believed that 3DP techniques can be classified into main two categories with laser and electron beam input systems [92]. The representative technologies are selective laser melting (SLM) and electron beam melting (EBM). The processes of SLM is also known as laser beam melting (LBM), direct metal laser Sintering (DMLS), LaserCUSING, or laser metal fusion (LMF) [92], as shown in Figure 2 [93].

The implant surface roughness fabricated using SLM technology (arithmetical mean roughness (Ra): 5–20 µm)) is smoother than the EBM (Ra: 20–50 µm) counterpart, because of its smaller laser spot size and thinner layer thickness (30–50 vs. 50–70 µm), smaller powder size (average diameter 30–50 vs. 60–80 µm), and lower energy input [94,95]. Many reports have proved that titanium and its alloys prepared using SLM and EBM methods improved the osseointegration [96]. Nevertheless, a common standard for the optimum roughness has not been introduced yet. A study [97] has confirmed that the osseointegration of titanium implant surfaces can be enhanced by achieving the roughness range from 0.5 to 2.0 µm, thus, it is necessary to do further surface modifications in order to improve surface roughness.

The cytotoxicity of Ti-6Al-4V made using SLM and EBM approaches was evaluated with fibroblasts, and it was seen that there was not any significant difference between values in comparison to the negative control group [98]. The almost same results were observed [99] in the cell proliferation of Ti-6Al-4V treated using SLM and EBM with mesenchymal stromal cells (MSC). In another study [88], different basic structures (cubic, diagonal, pyramidal) for Ti-6Al-4V scaffolds are produced using SLM and EBM. Under static conditions, human primary osteoblasts were cultured on the samples. The cell activity and matrix production in both of the two groups increased (no significant difference). The collagen type 1 in Ti-6Al-4V SLM and EBM scaffold specimens with 700 µm pore size and 51% porosity revealed a remarkable increase during osteoblast differentiation [88].

There is a crucial evaluation criterion, BIC percentage; a higher BIC percentage means a greater bone ingrowth level. Experimental studies [79,88,100] reported that the pore size and porosity of SLM were 250–800 µm and 63%, and the pore size and porosity of EBM were 350–1400 µm and 49%, respectively. In the Ti-6Al-4V specimens, BIC was not observed, and there was also no sign of histological differences in the femoral condyle of goats after four weeks of implantation between the two groups. After implanting for 15 weeks, both of the two groups were intimately connected to the host bone, and the histomorphometry results showed that the BIC value of the EBM specimen group was higher than that of the SLM [101]. 

Structural characteristics of scaffold have vital effects on the mechanical properties and biological performance. Several studies [100,102,103] have revealed that the different architectures with different pore sizes (100–1000 μm), porosity (30–80%), and pore shapes promote the initial osseointegration period. However, there is a controversy about the optimal structure. Table 1 summarizes the research literature and lists the influencing factors like material, technology, scaffold, and biological performance.

In summary, 3DP technology is used to produce biomedical metal implants with complex shapes, which promise good prospects for their clinical application in the future. The standard regulatory guidelines for additive manufactured medical devices are a prerequisite to further medical implantation. Meanwhile, the reliability and repeatability of stable physicochemical properties, biological characteristics, security, and its specifications are necessary. Especially, surface modification in the 3DP implants is an indispensable concept in order to attain high osseointegration performances. Insufficient bone formation, vascularization, contiguous infection, and implant durability are still the main challenges. Lastly, the excessive production cost of the 3DP technique limits its further development and large-scale application.

#### 2.1.2. Laser Surface Texturing

Nowadays, the bonding strength between bone and titanium implants can be increased using an LST technique [60,77,113]. These LST techniques have revealed a great potential to optimize the surface properties of biomedical implants through forming periodic textured patterns [80]. LST technology makes good use of thermal and photonic effects. The mechanism of LST is mainly based on ablation or vaporization. Accordingly, the effects of thermal conduction and fluid dynamics must be clarified during the process. There is no doubt that the processing efficiency and surface performance directly depend on the processing variables [77]. Thus, it must be pointed out that the topography and physicochemical and biological properties can be optimized according to different implant positioning. While some of the advantages of LST are obvious [114], for example, on the one hand, it is known as an environmentally friendly technology, on the other hand, it can modify the implant surfaces in a wide span range from macro-scale to nano-scale without any need for direct contact and it is free of any contaminations. Furthermore, the material surface treatment with this method is an automated procedure and can be used in complex shaped samples. In addition, the other advantage of the LST approach is its flexibility; it is also a non-contact procedure with high controllability and reproducibility. The process has a lower cost and higher efficiency compared to others, and it is suitable for automation and on-line monitoring. Hence, it can be utilized in the industrial applications instead of the ultrafast (femto/pico-second) laser [114]. However, some limitations and problems remain unsolved, such as the interaction between the laser beam with the material, making it difficult to be theoretically analyzed [77], and unfortunately, most of the previous studies focused on its empirical aspects. 

To enhance the wear resistance of Ti-6Al-4V surfaces, Kümmel et al. [115] produced a linear channel (width: 30 µm, depth: 10 µm) with a semicircular cross-section by means of LST processing. The wear volume of the LST samples were 16 times lower than non-textured reference samples (1.6 × 10^7^ µm^3^ VS. 0.1 × 10^7^ µm^3^). Patel et al. [116] fabricated different densities, shapes, and directions pillars of textures on Ti-6Al-4V using an LST method and found that the contact angle value was reduced by increasing the size of the micro-pillars from 30 × 30 µm to 100 × 300 µm. Texture size and orientation not only optimize the physical and chemical properties but also improve the biological performance of the implants [117,118,119,120,121,122,123,124,125]. Chen and Mwenifumbo et al. [117,118] proved that cell orientation and cell adhesion are improved when the width of the grooves is 11 μm and the depth is 10 μm. Cell adhesion strength tests have indicated that the highest cell retention was seen on the linear textured surfaces with 20 μm intervals. In addition, a linear pattern texture presented a higher rate of cell retention than the waved pattern textures [119,120]. Furthermore, the interaction mechanism of the grooves and cell proliferation was demonstrated in detail. Chen et al. [121] displayed the enhanced cell adhesion in micro-grooved surfaces because of the interaction between the focal adhesions and extracellular matrix (ECM) proteins. Brånemark et al. [122] showed that the micro-scale and nano-scale topography or surface oxides formation using laser treatment increased the bone-implant biocompatibility. Soboyejo et al. [120] reported that the MC3T3 cells maintained the contact guidance and aligned along the microgrooves. It was further explained that reducing longitudinal groove intervals leads to an increment in the cell contact guidance [123,124,125]. A previous study [126] demonstrated that the density of MC3T3-E1 cells dropped in the textured surface, especially in dimple textured surfaces. XTT assay showed the results of the cell viability of MC3T3-E1 fibroblast cells after 24 h. The result showed no toxic effect and good cell viability in the LST group, as shown in Figure 3a. More cells were attached to ridges and corners than on dimples of the textured surfaces, as shown in Figure 3d–f [126]. The same situation was observed in MG63 cells [109], and another study [127] indicated that the average roughness on the dimple feature (Ra = 3.5 µm) was higher than that of the linear feature of the surface (Ra = 2.7 µm). It has been shown that LST technology evidently reduces the adhesion of Staphylococcus aureus (*S. aureus*) bacteria and biofilm formation hence decreases the risk of implant-associated infections [128]. Biofilms are multi-species communities of microbial cells located on the extracellular polymeric matrix, quorum sensing communication, and offer nutrients for bacteria. Furthermore, an extracellular polymeric matrix prevents the operation of antibiotics on bacteria actually, and it acts as a biological barrier since the biofilm receives a positive response from the immune system [129,130,131,132].

The advantage of LST technology mainly involves the capability of hierarchically controlling the surface texture (for instance by producing pits, grooves, pillars, ablation tracks, ripples, and columns), array pitch, depth, and other parameters to further change the surface roughness and improve the material’s abrasion resistance, contact angle, biological properties (such as cell adhesion and biocompatibility, reduction of *S. aureus* adhesion) and ultimately improving antimicrobial and rapid bone integration.

### 2.2. Micro-Scale Treatment

#### 2.2.1. Grit-Blasting

After the processing of titanium samples to their final shape, usually, further surface treatment is required in order to roughen the surface, such as grit-blasting [1,11]. From long ago to the present day, grit-blasting has been an irreplaceable technology in surface treatment in which the hard ceramic particles are ejected by compressed air at a high velocity through a nozzle. The surface roughness mainly depends on the size of the ceramic particles, ranging from 110 to 250 μm. The ceramic particles should have some characterizations such as stability and biocompatibility, and they should also not affect the ingrowth of bone cells on titanium implants. Nevertheless, some entrapped abrasive particles are always found on the implant surfaces. These abrasive particles are usually of aluminum oxide (Al_2_O_3_), silicon oxide (SiO_2_), titanium oxide (TiO_2_), and calcium phosphate composition [133,134]. Al_2_O_3_ is often applied as the blasting material and it can be dissolved in acid. Blasting material particles are very hard to remove; even after ultrasonic cleaning, acid-etching, and sterilization they can be found on the sample surface. Evidently, these abrasive particles release in the peri-implant region and interfere with the osseointegration procedure. Furthermore, there is a possibility that these particles have a role in the reduction of corrosion resistance of implant surfaces in the body fluid environment [135]. 

TiO_2_ particles with an average size of 25 μm induce a medium roughness of about 1–2 μm. A study [136] revealed that the BIC value achieved by blasting the TiO_2_ particles remarkably is higher than the machined blasting surfaces. Meanwhile, current studies [68,137] have proved the BIC enhancement using TiO_2_ grit-blasting. After ten years of clinical implantation, studies [6,138] have shown a high clinical success rate of TiO_2_ grit-blasted titanium implant surfaces. Clinical studies [139,140] have reported higher marginal bone levels and survival rates after TiO_2_ grit-blasting than machined implants. A study [141] has revealed that when using grit-blasting with TiO_2_ or Al_2_O_3_ particles, the BIC does not show any significant differences. However, a TiO_2_ grit-blasted implant has increased mechanical fixation in comparison to smooth titanium surfaces. In addition, the torque force increases with the increment of surface roughness [142].

Calcium phosphate with its excellent compatibility, bioactivity, and biodegradability is also used as blasting media in titanium implant surfaces. Based on the crystalline type, calcium phosphates can be divided into α-tricalcium phosphate (α-TCP) and β-tricalcium phosphate (β-TCP). β-TCP and HA are identified as effective blasting materials, as these materials produce a clean and uniform texture on the titanium implant surface. The BIC value of calcium phosphate-treated surfaces is higher than that of machined surfaces [143,144]. Experimental studies have proved the BIC of a calcium phosphate-blasted surface was similar to other blasting materials during osseointegration.

A clinical study [145] reported the reliability of the secondary fixation by osseointegration in a straight standard grit-blasting titanium alloy used in non-anatomical implants. One hundred and ninety-eight Alloclassic™ total hip arthroplasties were performed in 179 patients, with a mean age of 66 years old (22–85), including 105 with proximal HA coating and 93 with the original grit-blast coating. The standard grit-blasted implant and HA coated standard grit-blasted implant are shown in Figure 4a,b [145]. The HA coating reduced the possible proximal fibrous encapsulation considering that the HA coating did not change the clinical results. Figure 4c [145] shows a straight Alloclassic™ THA (total hip arthroplasty) without HA coating implant in a 57-year-old female patient with a fracture of the femoral neck after removal of immediate postoperative control fixation hardware. The radiographic results after 23 years and 3 months of follow-up at the age of 80 years old and 5 months showed successful osseointegration, as shown in Figure 4d. These studies confirm that roughening the titanium implant surfaces increases bone-to-implant mechanical fixation but not its biological fixation.

#### 2.2.2. Acid-Etching

Acid-etching is defined as a procedure of roughening titanium implant surfaces with strong acid solutions including hydrochloric acid (HCl), nitric acid (HNO_3_), sulfuric acid (H_2_SO_4_), hydrofluoric acid (HF), and other combined acid solutions. In some cases, the purpose of acid-etching processing is to remove the blasting residual particles remaining from the previous grit-blasting processes on the implant surfaces. Acid-etching usually fabricates the structure of the micro-pits with pit sizes in the range 0.5–2 μm [146,147]. The micro-pits and spike-like peaks lead to 1–2.7 μm average roughness on the material surfaces. Specific roughness depends on acid types, acid concentration, reaction temperature, and reaction time. A common acid-etching procedure is as follows: the implants are immersed in an acid solution for one hour with ultrasonic vibration at 60–100 °C. Then, the produced oxidized films on a titanium surface are dissolved in acid solution [61,148,149,150]. However, acid-etching has a negative effect on mechanical performance. The procedure might result in hydrogen embrittlement in the titanium implants, meanwhile cracks produced on the surface possibly weaken the fatigue resistance of the titanium [151]. In fact, some studies confirm the absorption of hydrogen in titanium and the release of some amount of it in the liquid body environment. In addition, hydrogen embrittlement is related to the formation of brittle hybrid phases. This phenomenon is also associated with the fracture mechanisms of titanium implants [151]. 

Surfaces treated using dual acid-etching (24% HF + HCl/H_2_SO_4_) can accelerate bone ingrowth, keeping its long-term success rate [152]. An experimental study [153] has reported that acid-etching surfaces have the capability to strengthen osseointegration, generally achieved by the attachment of fibrin and osteogenic cells around the implant surface. And woven bone with thin trabeculae covering the implant has been observed [154]. An experimental study [155] raised a presumption that dual acid-etching surfaces could attach to the fibrin scaffold and promote the adhesion of osteogenic cells. Studies [156,157] have demonstrated that dual acid-etching surfaces show higher BIC values and less bone absorption than machined surfaces. In recent decades, the acid-etching approach has been developed to enhance cell adhesion and bone formation. Another process involvess fluoride solution, which is used to modify the titanium surface, as titanium can react with fluoride ions, while forming soluble TiF_4_ species. This chemical treatment roughens the titanium surface and introduces TiF_4_ on the surface, which in turn promotes rapid bone ingrowth [158]. Another study [159] has reported that surfaces during HF treatment act in favor of osteoblastic differentiation when compared with the control group. In addition, fluoridated rough implants sustain higher strength and torque removal in comparison with control samples [160]. As is shown in Figure 5 [161], the morphology of an acid-etching surface of a commercially available implant is uniform with pits of approximately 3 μm in width [161,162]. The acid-etching method shows enormous potential in the improvement of bone-to-implant fixation due to an increase of bioactivity on the implant surface.

#### 2.2.3. Sand-Blasted, Large-Grit, and Acid-Etched

In general, surface treatment by combining grit-blasting with acid-etching procedures is defined as SLA. Experimental studies [163,164,165,166,167] reported that SLA treated surfaces are beneficial, with increased biocompatibility in early bone formation stage and also in cell differentiation. The clinical success rate of SLA has achieved about 95% [168,169,170]. After SLA treatment, the topography of titanium implant surfaces provides positive effects on the activation of blood platelets and cell migration. Several experimental studies [9,169,171,172,173] demonstrated that the hydrophilicity of implant surfaces can further shorten the osseointegration process. In order to improve hydrophilicity, titanium implants are treated using SLA, and they are immersed in isotonic solution at low pH to produce a super-hydrophilic titanium surface. The approach is usually done by a commercial brand named SLActive [174]. The chemical stability can be fixed by the combination of acid-etching and conditioning it in an isotonic solution. The implant’s surface turns super-hydrophilic due to the application of chemical surface modification. By utilization of an SLA procedure in isotonic solutions, some spike-like nanofeatures can be produced on the surface of titanium [148,163,173]. Furthermore, it was seen [173] that the bonding strength increased between the nanostructured surface and bone tissue, measured by mechanical pulling out tests in rabbit tibia for eight weeks. Nanostructure features with further acceleration of the bone healing process are able to enhance protein adsorption, platelet aggregation, and macrophage adhesion [15,163,169,173,175,176,177]. An experimental study [178] revealed the up-regulation of pro-osteogenic cell signaling pathways and osteocalcin on ultra-hydrophilic titanium surfaces. Compared with acid-etching surfaces, the super-hydrophilic surface can increase BIC in 2–4 weeks [179,180,181,182,183,184]. The average BIC on SLA surfaces showed to be in the range of 67–81% for 6 months. A 10-year follow-up study [185] of marginal bone loss demonstrated that the clinical success rate reached 95.1% for acid-etching implant surfaces. Other studies [10,186] of immediate provisional restorations on implant surfaces have reported a clinical success rate of about 100% on super-hydrophilic implant surfaces with positive aesthetic outcomes. Zhang et al. [167] demonstrated the osteogenic performance of SLA and 3DA (3DP and acid-etching) implants in the femoral condyle of SD rats for 3 and 6 weeks, as is shown in Figure 6 [167]. The BIC of an SLA implant as higher than that of a 3DA implant (in Figure 6a,b) [167]), thus, SLA processing still cannot be replaced. Micro-scale and nano-scale modification have revealed a positive effect on osteogenic cell growth because they produce a hierarchical structure by imitating the skeleton.

### 2.3. Nano-Scale Treatment

#### 2.3.1. Plasma-Spraying

For several decades, plasma-spraying, as a safe and reliable nano-scale coating technology, has been used for roughening implant surfaces. Plasma-spraying equipment consists of a DC electrical power source, gas flow control, a water-cooling system, and a powder feeder. Plasma spraying technology is a physical method, which involves spraying melted coating material onto Ti substrate surfaces using a direct current arc plasma gun, producing a 30-μm thick coating. Actually, the optimal thickness of the film is approximately 50 μm [67] and the average roughness of the coating is approximately 7 μm, and it also increases the implant surface area [67]. 

A study [187] has reported that a three-dimensional topography formation increased the mechanical interlock and tensile strength between bone and implant surfaces. The bone-to-implant interface was produced faster after plasma-spraying treatment compared to that of smooth surfaces. Nevertheless, titanium particles are observed in the peri-implant region [188]. The observed titanium wear particles are from the bone-to-implant interface, scattering among the organs in the minipigs implant experiment [188]. The released metallic ions are the product of dissolution or wear processes. In this regard, the local and systemic carcinogenic potential effects may attract people’s attention and leads to some limitations in its clinical application [189]. No clinical differences between SLA and plasma-spraying methods in the interface of titanium implants were reported in a study by Loughlin [190]. Another study [191] showed that the BIC on the plasma-sprayed surface is lower than on the plasma-sprayed HA coating surface. 

A large number of studies [81,192,193,194,195,196] have reported composite materials coatings modification using plasma-spraying treatment on titanium implant surfaces. Li et al. [192] fabricated nano-TiO_2_/Ag and nano-TiO_2_ coatings using a plasma-spraying technique on titanium substrates to improve the bioactive and antibacterial properties. From water contact angle and MG-63 cell adhesion and proliferation tests, there were no significant differences between nano-TiO_2_/Ag and nano-TiO_2_ samples. However, in the samples containing Ag particles, the percent reduction of Escherichia coli reached approximately 100% after 24 h, and the loaded Ag particles did not show obvious osteo-toxicity. Ke et al. [194] produced the HA layer using laser engineered net shaping (LENSTM) on Ti-6Al-4V substrates, and then prepared HA/MgO/Ag_2_O coating using plasma-spraying in order to enhance the strength of the adhesive bond between the coating and substrates, as shown in Figure 7 [194]. Compared with just plasma-spraying coating condition, LENS^TM^ and plasma-spraying procedures increased the bond strength from 26 ± 2 MPa to 39 ± 4 MPa. Additionally, the Ag ions release amount reduced to 70% due to crystallization enhancement by the LENS^TM^ HA layer. In vitro human osteoblast cell culture assays indicated that Ag_2_O (2 wt%) was a quite safe coating since an antibacterial characteristic was observed against E. coli and S. aureus in Ag_2_O coatings. Another investigation [195] has reported improved wettability after plasma-spraying treatment. The results revealed that unheated treated HA-ZrO_2_ and HA-TiO_2_ coating modified by plasma-spraying showed better hydrophilicity than the heat-treated condition, and the water contact angle was 25° and 35°, respectively. It is worth mentioning that both ZrO_2_ and heat treatment can enhance the hardness of material surfaces [196]. In a Sprague-Dawley rats model experiment for five weeks, the rate of bone mineralization of plasma-spraying ZnO(0.25 wt%)/SiO_2_ (0.5 wt%)/Ag_2_O(2.0 wt%)-HA composite coating was 32%, while it was about 11% in the plasma-spraying HA coating group [197]. Utilization of TiO_2_, Ag_2_O, ZrO_2_, ZnO, and SiO_2_ in a suitable content is beneficial for the antibacterial property, hardness, osteo-conduction, and early bone formation [193].

In fact, the corresponding standards about the plasma-spraying treatment on titanium implants in clinical applications are briefly reported in this review paper. However, this technology still has enormous potential to develop in the future.

#### 2.3.2. Anodization

Anodization technology is a mature technology to change the roughness and topographic features on the surface of titanium with many influencing variables, for instance, oxidation duration, oxidation voltage, electrolyte solution type, electrolyte solution concentration, and the subsequent heat treatment process. Nanopores and nanotubes can be induced by constant potential anodization in different acid solutions (e.g., H_2_SO_4_, HF, H_3_PO_4_, HNO_3_) for various time spans [11,198]. A uniform oxide layer forms on the titanium surface with a thickness of about a few hundred nanometers up to a few microns [199]. The anodic oxide film is formed by the charging of the double electric layer at the metal-electrolyte interface. The mechanism is dissolution of oxide film assisted by the electric field and it is enhanced by temperature, involving the formation of a soluble salt containing the metal cation and an anion in the electrolytic bath. After establishing a stable potential, the current gradually decreases due to either a decrease in Ti^3+^ in the membrane layer or an increase in the integrity of the membrane layer, which can lead to a significant increase in resistance, resulting in a reduced current [200]. Compared with machined surfaces, anodized surfaces enhanced the bone response in biomechanical and histomorphometric experiments [201]. The anodized preparation process and TiO_2_ nanotubes array are shown in Figure 8a,b. In Figure 8c, CaO is observed on the surface of NT-RP-Ca/P. The potentiodynamic polarization curves observed in Figure 8d show that samples containing nanotube (NT and NT-RP-Ca) exhibited a passive region extending over a wide potential range when compared to Ti surfaces. Bone-like structured TiO_2_ nanotubes displayed superior corrosion resistance ability. In addition, bone-like structured TiO_2_ nanotubes enriched with calcium and phosphorous have enhanced osteoblastic cell functions with MG-63 cells, as is shown in Figure 8e [198]. In another study [202], the clinical success rate of anodized implants are reported to be higher than that of machined titanium surfaces. There are two mechanisms to explain the osseointegration: mechanical interlocking and biochemical bonding observed between implant material and bone [66,203]. Among the many metal and salt ions (Ti, Mg, P, Ca, S) [204,205], the incorporation of Mg ions is the best way to remove the torque value [66].

The anodized studies in references [198,206] possess nano-scale surfaces, enabling them to load and deliver multifunctional molecules and growth factors to accelerate early bone integration. The wall thickness, diameter, and length of nanotubes directly depend on the anodization parameters such as oxide temperature, voltage, time, and electrolyte concentration [199]. Nanotubes increase the contact surface area resulting in an increase in wettability and adsorption of proteins and ions [198,207,208,209]. Loading antibacterial ions on nanotubes can prevent biofilm formation and reduce the bacteria in the peri-implant region to avoid early failure of the implant [210,211,212]. The length range of anodized nanotube array lies between 7 and 10 μm. The inner diameter range and the external diameter range are 20–100 nm and 30–110 nm, respectively [198,206]. The nanotube spacing is suitable for the transformation of waste and nutrients [213]. The TiO_2_ nanotube size can be adjusted by changing the anodization parameters in order to achieve a similar size as the skeleton. The diameter of cortical bone is reported to be in the range of 10–500 μm, while the diameter of the cancellous bone is 0.2–1 mm [14,199,214]. In addition, nanotube arrays try to simulate the size and arrangement of collagen fibrils in the bone tissue [215]. Several experimental studies [216,217] indicated that the length of TiO_2_ nanotubes has an influence on biocompatibility, while their diameter has a critical effect on cell adhesion and proliferation. The best osteoconductivity was reported in for 70 nm diameter nanotubes, meanwhile, 80 nm diameter nanotubes also showed improved proliferation and differentiation behavior [199,218,219]. Additionally, the extra annealing (600 °C 3 h) in the heat treatment stage seems to be beneficial for the best wettability behavior with a 62° water contact angle for improved cellular response [220]. There are three crystalline phases of TiO_2_ including anatase, rutile, and brookite. Anatase forms with annealing at 400–600 °C for 2–3 h [221]. Rutile begins to form with annealing at more than 700 °C for 2–3 h [222]. Furthermore, the anatase crystallization of TiO_2_ nanotubes enhances the hydrophilicity of the annealed surfaces resulting in a rise in protein adsorption [223].

In brief, nano-scale feature formation on titanium can improve protein adsorption, osteoblastic cell adhesion and proliferation, and the healing rate of the implant periphery zone. The anodization procedure produces a uniform and regular topography; in particular, TiO_2_ nanotube arrays are also developed as a drug loading system to deliver corresponding drugs. The advantage of this system is that the delivered multifunctional drugs can be released in the predetermined time span and then can be released into the interface of titanium implants [224]. TiO_2_ nanotubes are beneficial for utilization in drug delivery systems. The tube length, diameter, and phase are adjustable according to the desired demands. In view of common electrochemical approaches to fabricate micro/nanopores and nanotubes on implant surfaces in electrolyte solutions with a settled voltage and time, the production of standard implants seems to be feasible from an industrial perspective. Considering the present studies about nanotube fabrication on titanium implants, the special design and identification of the modern nanotube-modified implants depend on their clinical benefits, the demands of patients, and the interests of the producers.

## 3. Conclusions

There are a large number of surface treatment approaches commercially available for producing titanium implant surfaces. Nowadays, surface modification techniques (for instance, 3DP, grit-blasting, acid-etching, plasma-spraying, and anodization) have proven clinical efficacy. Reaching to a favorable surface morphology after surface modification plays a vital role in enhancing early osseointegration. In particular, multi-scale combined topography (such as macro-scale, micro-scale, and nano-scale) can shorten the phase of bone ingrowth. Blood platelet activation, protein adsorption, three-dimensional fibrin clot cross-linking, osteogenic cell migration, collagen deposition, and bone matrix formation are the main factors affecting the osseointegration process. Wettability, roughness, and chemical composition are the bridges connecting the physicochemical properties and biological properties of titanium alloy surfaces. It’s worth noting that there are still no strict requirements and qualified standards for implant surface morphology design. In addition, a large amount of pre-clinical and clinical experiments need to be done to further ensure the security and reliability of implants using new technologies. In addition, the high cost is another limitation that creates a lot of difficulties in the clinical validation stage of implant design. The future modern implants should satisfy the following characteristics: biomimetic and standardized properties, slow rate of material release into the body environment, and low cost. Multi-scale surface treated implants show considerable potential in order to design modern implant materials with enhanced properties.

## Figures and Tables

**Figure 1 nanomaterials-10-01244-f001:**
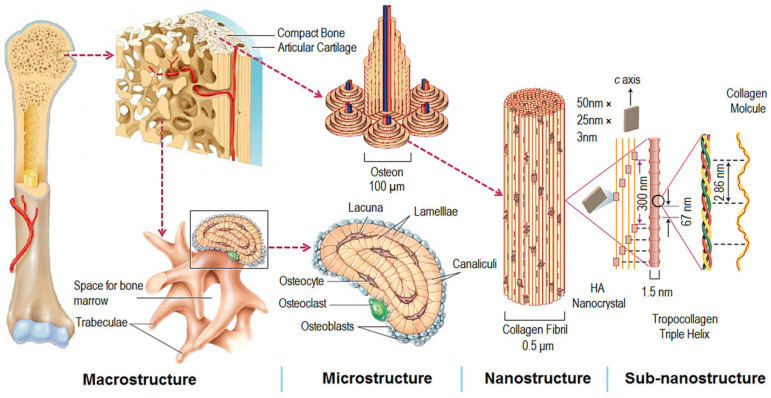
Structure of skeleton: descending hierarchical macro- to nano-scale structures of natural bone. (Reproduced with permission from [62]. Copyright Elsevier, 2016).

**Figure 2 nanomaterials-10-01244-f002:**
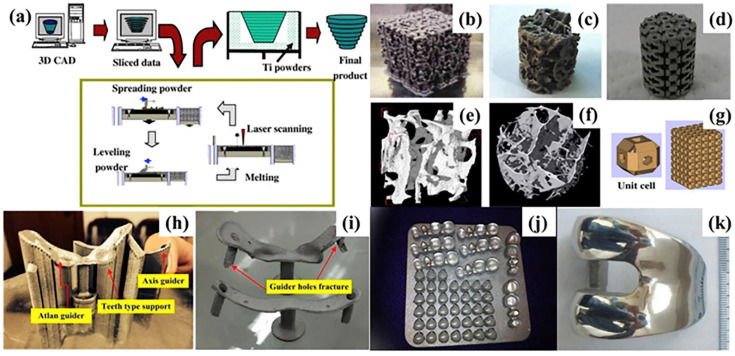
(**a**) Schematics of a laser beam melting (LBM) machine; (**b**–**d**) porous structures; (**e**–**f**) micro-CT images of human cancellous bones; (**g**) stacked hollow cubes; (**h**–**i**) the surgical template in pre-blasting and after blasting condition; (**j**) the dental restorations; (**k**) a personalized femoral component. (Reproduced with permission from [93]. Copyright Elsevier, 2019).

**Figure 3 nanomaterials-10-01244-f003:**
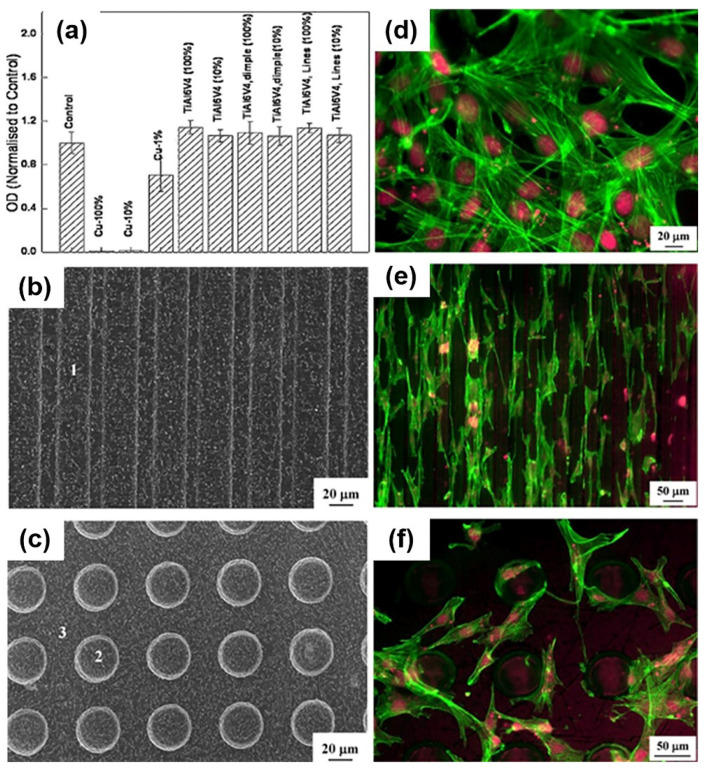
(**a**) XTT (Dimethoxazole yellow) results of cell viability of MC3T3-E1 fibroblast cells after 24 h in contact with the extracts in the as-received and laser textured surface; (**b**,**c**) SEM of the surface of linear geometry and dimple geometry; (**d**–**f**) fluorescent micrographs of the as-received, line geometry and dimple geometry showing the attachment of MC3T3-E1 cells. (Reproduced with permission from [126]. Copyright Elsevier, 2015).

**Figure 4 nanomaterials-10-01244-f004:**
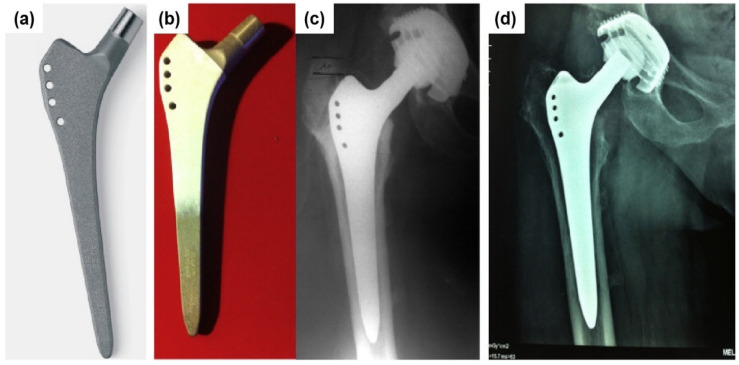
(**a**) Standard version (grit-blasting); (**b**) HA coating on standard version; (**c**) straight Alloclassic™ THA in a 57-year-old female patient for nonunion of a fracture of the femoral neck after removing fixation hardware: immediate postoperative control; (**d**) radiographic result after 23 years and 3 months of follow-up at the age of 80 years old and 5 months. (Reproduced with permission from [145]. Copyright Elsevier, 2014).

**Figure 5 nanomaterials-10-01244-f005:**
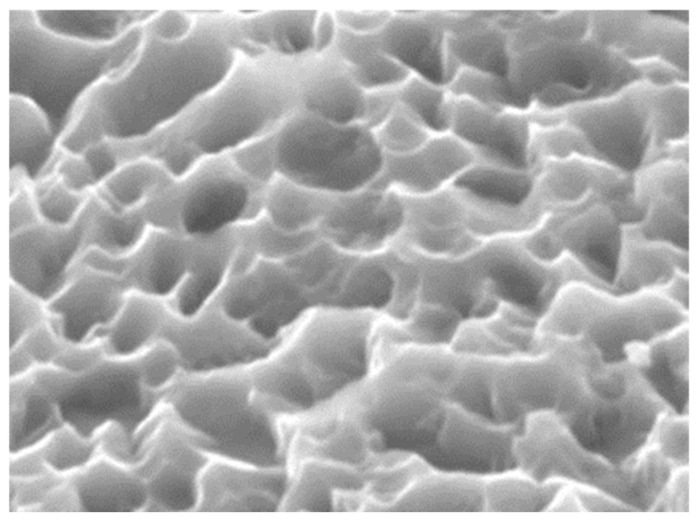
Typical dental implant surface morphology using acid-etching. (Reproduced with permission from [161]. Copyright Elsevier, 2013).

**Figure 6 nanomaterials-10-01244-f006:**
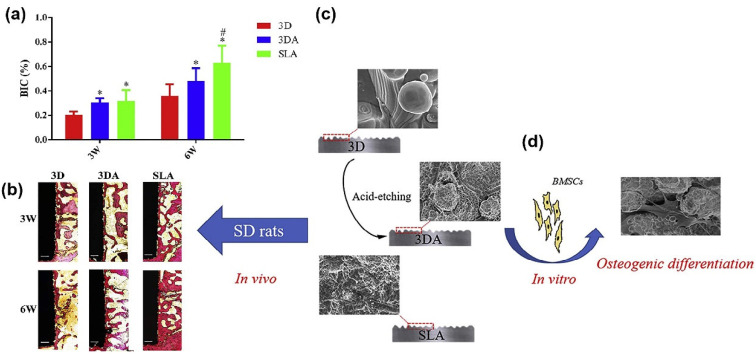
(**a**) Representative histological images of 3D, 3DA, and SLA implants after implantation for 3 and 6 weeks, respectively (scale bar = 200 μm); (**b**) quantification of BIC percentages on implant surfaces; (**c**) SEM of 3D, 3DA, and SLA surfaces; (**d**) cell morphology on the 3DA surface after culturing of bone marrow stromal cells (BMSCs) for 24 h observed using SEM. (Reproduced with permission from [167]. Copyright Elsevier, 2020).

**Figure 7 nanomaterials-10-01244-f007:**
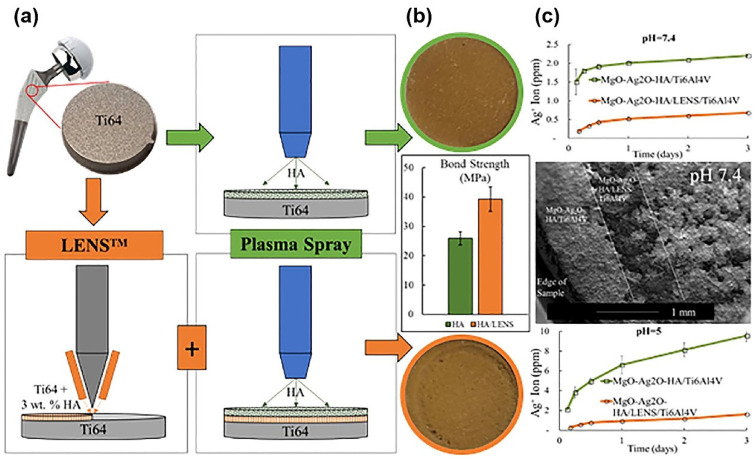
(**a**) Schematics during laser engineered net shaping (LENS^TM^) and plasma-spraying treatments; (**b**) bond strength between HA and HA/LENS coatings and substrates; (**c**) accumulative Ag^+^ release in MgO-Ag_2_O-HA/Ti-6Al-4V and MgO-Ag_2_O-HA/LENS/Ti-6Al-4V group. (Reproduced with permission from [194]. Copyright Elsevier, 2019).

**Figure 8 nanomaterials-10-01244-f008:**
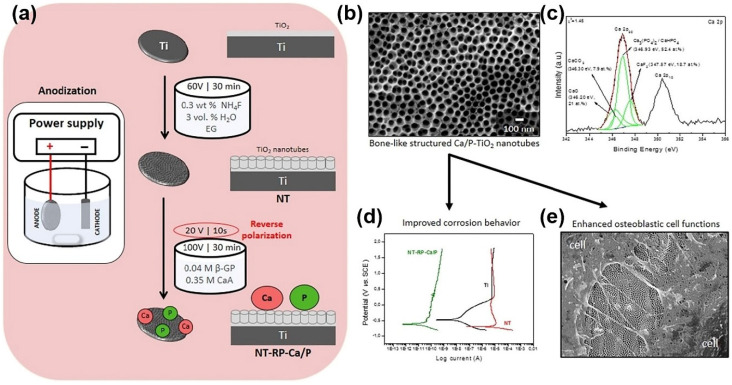
(**a**) Schematics during anodization; (**b**) FESEM micrographs showing the morphology of the highly ordered TiO_2_ nanotubes present on NT-Ca/P; (**c**) high resolution XPS spectra of deconvoluted Ca 2p; (**d**) potentiodynamic polarization curves of Ti, NT, NT-Ca/P, and NT-RP-Ca/P samples immersed at 37 °C; (**e**) FESEM micrographs of MG-63 cells cultured on NT-RP-Ca/P surfaces after one day of incubation. (Reproduced with permission from [198]. Copyright Elsevier, 2016).

**Table 1 nanomaterials-10-01244-t001:** The important parameters for 3DP architectured implant preparation, in vivo studies.

3DP Material	3DP Method	Pore Size (μm) ^(a)^	Porosity (%)	Pore Shape	Animal Model	Time ^(b)^	Result	Ref.
Ti	SLM	300, 600, 900	61.6 66.4 64	diamond lattice with hexagonal pore throat shape	Rabbit femur	8W	P600 implant is a suitable porous structure.	[104]
Ti-6Al-4V (grade 5)	EBM	500–700	65–70	\	Sheep vertebra	26W	A higher degree of osseointegration was observed inside the porous structure than in that of the dense group.	[105]
Ti-6Al-4V	SLM	600	70	\	Beagle tibia	12W	The volume of regenerated bone increased with increase of the implantation time (from 11.89% at 4 weeks to 15.85% at 12 weeks), which was better than the Ta group.	[106]
Ti-6Al-4V	EBM	710	68	\	Sheep vertebral	6M	Ti cages demonstrated better osteointegration with surrounding bone tissue than PEEK cages.	[107]
Ti-6Al-4V	SLM	900, 1200	84, 54	diamond lattice	Sheep tibia	2M	900 μm lattice cell size was more favorable to bone ingrowth.	[108]
Ti-6Al-4V	EBM	450	61.3	\	Domestic pig skull	2M	The bone volume inside the implants reached almost 46%. BIC was achieved at 5.96%.	[109]
Ti-6Al-4V	3DP	300–500, 200–600, 100–700	49.53	\	Bama mini pig tibia	5W	The bone volume/total volume was 12.71–3.556%, 11.99–3.581%, and 12.84–3.874%, respectively.	[110]
Ti-6Al-4V	EBM	\	\	\	Rabbit femur	2W	The implants with an EBM screw had a higher BIC ratio (≈35%) than those with the machine-implanted screw (≈5%).	[111]
Ti-6Al-4V	EBM	500, 640, 800, 1000	65, 70, 67	diamond lattice	Rabbit distal femur	12W	Pore size of 500–800 μm showed more favorable histological bone ingrowth than 1000 μm.	[100]
Ti-6Al-4V	SLM	500, 600, 700	60, 70	octahedral	Rat Sprague-Dawley	12W	Pore size of 500 μm and porosity of 60% had the highest BV/TV and hence the best bone ingrowth.	[112]

(a) Pore sizes were presented as designed (measured). (b) W and M mean weeks and months.

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
