# Peer review of "Multi-Scale Surface Treatments of Titanium Implants for Rapid Osseointegration: A Review"

_nanomaterials, 2020, doi:10.3390/nano10061244_

Round 1
Reviewer 1 Report
A strong improvement in English is mandatoty. There are several typos errors, and some lack of coherence in the use of terms: 1) both acidetching and acid-etching were used
Some comment are hereafter listed.
Introduction:
- the author stated “Dental implant shapes have transformed from hexagonal to conical connections and cylindrical to conical design up to now and they usually designed as rough titanium surfaces”. A justification is necessary
- line 12: this sentence should be written in a better English
- line 21: add a space after the dot
- line 29: leads instead of lead
- line 36 and 45: typos
Section 1.1 line 18-19: in this sentence the author wants to talk about a regular SLA process and a modified one without telling the entity of the modifications.
Section 1.2 line 12-14-19: typos
Section 2.1.1: SLS and SLM are two different techniques. PLs justify and comment on it
Section 2.1.2:
- the author should provide a correlation between technological parameters and surface features
- line 27 to 44: this part is only a list of findings of other researchers without providing comparison and personal opinion of the author regarding the effect and how to improve the surface quality
Section 2.2.1: Alloclassic needs the trademark
Section 2.3.1: the technique is poorly described
Section 2.3.2:
- the mechanism of porosity formation is poorly described. It is a mechanism of dissolution assisted by the electric field and enhanced by temperature, involving the formation of a soluble salt containing the metal cation and an anion contained in the electrolytic bath
- line 11: typos
- line 18-20: this period should be written in a better English
- line 29 and 59: typos
- anatase forms in a wide temperature range not only in correspondence of 450 °C
Table 1 and Table 2 should be redrawn and scaled according to the page size
Fig. 4: typos
Fig. 8: is poorly described for example no description of the potentiodynamic tests was provided
Reviewer 2 Report
In the manuscript entitled: “Multi-scale surface treatments of titanium implant for rapid osseointegration: A review”, the authors evaluated the advances in multi-scale surface technology of titanium implants in order to accelerate the osseointegration process. In addition, the several multi-scaled methods used for improving wettability, roughness and bioactivity of implant surfaces are reviewed.
In their in study, the authors evaluated macro-scale methods (e.g. 3D printing (3DP), laser surface texturing (LST)), micro-scale (such as grit-blasting, acid-etching, sandblasted, large-grit, and acid-etching-SLA) and nano-scale methods (for instance, plasma-spraying, anodization).
The authors concluded that a large number of in vitro and in vivo investigations are necessary in order to obtain in depth exploration of functional implant surfaces and their efficiency. The prospective implant surfaces should possess the controllable and standardized modern topographies with improved chemistry and bionic characteristics.
Major comments:
In general, the idea and innovation of this study, regards the analysis of titanium surfaes in dental field is interesting, because the role of dental implant survival characteristics are validated but further studies on this topic could be an innovative issue in this field could be open a creative matter of debate in literature by adding new information. Moreover, there are few reports in the literature that studied this interesting topic with this kind of study design.
The study was well conducted by the authors; However, there are some concerns to revise that are described below.
The introduction section resumes the existing knowledge regarding the important factor linked with osteointegration causes.
However, as the importance of the topic, the reviewer strongly recommends, before a further re-evaluation of the manuscript, to update the literature through read, discuss and cites in the references with great attention all of those recent interesting articles, that helps the authors to better introduce and discuss the role of periodontitis and gingival mediators useful for osteointegrations, such as ADMA, endothelin and vitamins: 1) Isola G, Alibrandi A, Currò M, Matarese M, Ricca S, Matarese G, Ientile R, Kocher T. Evaluation of salivary and serum ADMA levels in patients with periodontal and cardiovascular disease as subclinical marker of cardiovascular risk. J Periodontol. 2020 Jan 7. doi: 10.1002/JPER.19-0446. 2) Isola G, Polizzi A, Alibrandi A, Indelicato F, Ferlito S. 2) Analysis of Endothelin-1 Concentrations in Individuals with Periodontitis. Sci Rep. 2020 Feb 3;10(1):1652. doi: 10.1038/s41598-020-58585-4. 3) Isola G, Alibrandi A, Rapisarda E, Matarese G, Williams RC, Leonardi R. Association of vitamin D in patients with periodontitis: A cross-sectional study. J Periodontal Res. 2020 Mar 16. doi: 10.1111/jre.12746.
The authors should be better specified, at the end of the introduction section, the rational of the review and the aim of the review. In the material and methods section, should vetter underline the role of Acid-etching on titanium surfaces and the Anodization.
The discussion section appears well organized with the relevant paper that support the conclusions, even if the authors should better discuss the relationship between periodontitis and peri-implantitis mediated by the previous specified mediators. The conclusion should reinforce in light of the discussions.
In conclusion, I am sure that the authors are fine clinicians who achieve very nice results with their adopted protocol. However, this study, in my view does not in its current form satisfy a very high scientific requirement for publication in this journal and requests some revisions before a further evaluation of the manuscript.
Minor Comments:
Abstract:
- Better formulate the introduction section by better describing the aim of the study
Introduction:
- Please refer to major comments
Discussion
- Please add a specific sentence that clarifies the results obtained in the first part of the discussion
- Page 24 last paragraph: Please reorganize this paragraph that is not clear
Round 2
Reviewer 1 Report
the paper is now accepted
Reviewer 2 Report
In the R1 version of the manuscript entitled: “Multi-scale surface treatments of titanium implant for rapid osseointegration: A review” the authors followed all the issues suggested by the reviewer. Though the changes based on the reviewer comments, almost of the criticisms were carefully analysed and solved.
I have carefully evaluated all parts of the manuscript. I believe that the article, in this version, is now adequate for publication in this journal.